# Adaptability Analysis of the Evergreen Pioneer Tree Species *Schima superba* to Climate Change in Zhejiang Province

Chuping Wu [1,2], Jianzhong Fan [3], Yonghong Xu [3], Bo Jiang [1,2], Jiejie Jiao [1,2] and Liangjin Yao [1,2,*]

1    Zhejiang Academy of Forestry, Hangzhou 310023, China; wcp1117@hotmail.com (C.W.); jiangbof@126.com (B.J.); jjjjust@163.com (J.J.)
2    Zhejiang Hangzhou Urban Forest Ecosystem Research Station, Hangzhou 310023, China
3    Forestry Farm of Jiande, Hangzhou 310023, China; 18069810771@163.com (J.F.); xuyonghong305@163.com (Y.X.)
*    Correspondence: lj890caf@163.com

**Abstract:** In recent years, frequent global climate change has led to extreme weather events, such as high temperatures and droughts. Under the backdrop of climate change, the potential distribution zones of plants will undergo alterations. Therefore, it is necessary to predict the potential geographical distribution patterns of plants under climate change. *Schima superba*, a plant species with significant ecological and economic value, plays a crucial role in ecological restoration and maintaining environmental stability. Therefore, predicting potential changes in its suitable habitat in Zhejiang Province is significant. The MaxEnt model and combined data from 831 monitoring sites where *Schima superba* is distributed in Zhejiang Province with 12 selected bioclimatic variables were used to predict habitat suitability adaptability. We found that (1) the average AUC value of the MaxEnt model in repeated experiments was 0.804, with a standard deviation of 0.014, which indicates high reliability in predictions. (2) The total suitable habitat area for *Schima superba* in Zhejiang Province (suitability value > 0.05) is 87,600 km$^2$, with high-suitability, moderate-suitability, and low-suitability areas covering 29,400 km$^2$, 25,700 km$^2$, and 32,500 km$^2$, respectively. (3) Likewise, elevation, precipitation, and temperature are the dominant climatic variables that influence the distribution of *Schima superba*. *Schima superba* mainly occurs in areas with an elevation above 500 m and precipitation over 140 mm during the hottest season. The probability of *Schima superba* distribution reaches its peak at elevations between 1200 and 1400 m. Here, the precipitation ranges from 300 to 350 mm with high humidity, between 160 and 170 mm during the hottest season, and an annual temperature range between 28 and 31 °C. Therefore, our results indicate that climate change significantly affects the suitable habitat area of *Schima superba*. We also reveal the ecological characteristics and adaptation mechanisms of *Schima superba* in different geographical regions of Zhejiang Province. Future research should focus on the relationship between plant adaptation strategies and environmental changes, as well as applications in ecosystem protection and sustainable development, to promote the development and application of plant habitat adaptability research.

**Keywords:** adaptation analysis; climate change; habitat suitability zone; model evaluation

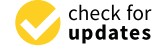



## 1. Introduction

In the context of climate change, the potential distribution of plants is expected to undergo alterations [1–3]. Therefore, it becomes imperative to predict the impact of climate change on the potential geographical distribution patterns of plants [2–4]. The habitat suitability of plants refers to their ability, either as individual populations or as species, to survive and reproduce in specific environments characterized by various biotic and abiotic factors, such as temperature, humidity, light availability, soil types, nutrient levels, salinity, and climate change [5,6]. This topic has consistently been one of the focal areas of research in plant ecology and physiology. Plant habitat suitability encompasses several

dimensions, including morphological and physiological traits, seed ecology, genetics, and gene regulation [7–9]. Plant adaptability refers to the ability of plants to respond to various environmental changes through a range of physiological, morphological, and biochemical mechanisms. These factors encompass temperature, moisture, light, soil, and various other aspects. By adapting to their environment, plants can enhance their survival and reproductive capabilities [10,11]. For instance, plants can adapt to varying humidity and light conditions by adjusting stomatal size and density [12]. They can cope with diverse soil types and nutrient levels by altering root system architecture and root exudation. Additionally, plants can combat environmental stressors by synthesizing specific protective compounds, such as antioxidants and antifreeze proteins. The significance of studying plant habitat suitability lies in gaining a deeper understanding of the survival strategies and adaptation mechanisms that plants employ under different environmental conditions. This knowledge aids in elucidating the distribution, competition, and evolutionary processes of plants within natural ecosystems [4,6]. Furthermore, given an escalation in global climate change, research on plant adaptability holds crucial practical implications that provide scientific foundations for agriculture, ecological restoration, and plant breeding. Through an in-depth exploration of plant habitat suitability, we can enhance our comprehension of the interactions between plants and their surroundings and contribute to the protection and sustainable utilization of plant resources by promoting ecological well-being and sustainable development in human societies [3,4,13].

The growth morphology, physiological responses, and reproductive strategies of plants exhibit varied responses and adaptive mechanisms to various types of environmental stressors, such as high temperatures, drought, salinity, and cold [13]. Recently, studies were able to simulate the potential distribution of bamboo forests in China under future climate scenarios based on climate variables and maximum entropy modeling [14]. Their results revealed that precipitation and temperature changes significantly influenced the potential distribution of bamboo forests. Additionally, the suitable growing area of bamboo forests in China increased initially and then decreased under low carbon emission (RCP4.5) and high carbon emission (RCP8.5) climate conditions, respectively. The growth range also contracted in the inland direction but expanded towards the southwest. This information provides a beneficial reference to dynamically monitor the spatial distribution and sustainable utilization of bamboo forests under future climate change conditions, which is crucial for the sustainable management of bamboo forests and the development of the bamboo industry [2,3,13,14].

Factors such as environmental selection and geographical isolation play pivotal roles in plant evolution and speciation [15]. Population differentiation, adaptive mutations, and natural selection significantly interact with various environmental factors, including soil conditions, water use efficiency, photoperiod adaptation, and climate change [14–16]. These factors also make essential contributions to species formation and species diversity. Consequently, the application of model-based predictions and decision-making methods in plant adaptation protection becomes increasingly crucial [16]. For example, many recent studies have explored the limitations on plant adaptability posed by climate change that encompass shifts in environmental conditions, lack of genetic diversity, and alterations in species interactions [3,17]. Specifically, they have discussed how models can be employed to forecast plant responses and adaptability to climate change that emphasize the importance of preserving genetic diversity, establishing protected area networks, and promoting the conservation of inter-species interactions [3,17]. Likewise, recommendations have also been proposed to enhance models and data [16,18]. The analysis of plant habitat suitability provides valuable insights into understanding the constraints on plant adaptability under environmental changes and formulating corresponding protection strategies.

*Schima superba*, an ancient angiosperm, belongs to the family Theaceae. It is an evergreen tall tree known for its rich biological characteristics and ecological functions that exhibit high ecological adaptability to its environment. Zhejiang Province boasts abundant *Schima superba* resources and acts as one of the crucial hotspots for biodiversity in

China [18]. However, due to environmental changes and human activities, the habitat adaptability of *Schima superba* faces certain challenges. Therefore, models that predict the potential distribution of *Schima superba* based on species distribution information and corresponding environmental variables (species distribution models, SDMs) would contribute to enhancing our understanding of the ecology and biogeography of *Schima superba*.

In terms of biological populations, the MaxEnt model can be used to predict species and habitat distributions, especially in data-limited situations. By utilizing known constraints between species and habitats, the MaxEnt model infers the probability distribution of species. Additionally, the MaxEnt model can assess a species' responsiveness to different environmental changes, which provides decision support for biodiversity conservation and ecosystem management [14]. Ultimately, this study uses *Schima superba* survey data and environmental factors from 831 fixed forest monitoring plots across 20 counties (cities) in Zhejiang Province to understand the ecological characteristics, distribution patterns, and the relationship between *Schima superba* and these environmental factors. Altogether, we look forward to using the MaxEnt model to thoroughly investigate the key factors influencing the geographical migration of *Schima superba* under future global climate change. The aim is to understand the constraints of climate variables on the potential geographical distribution of *Schima superba*, providing a scientific basis for the efficient utilization of its germplasm resources.

## 2. Research Area and Methodology

### 2.1. Study Area

Zhejiang Province has a land area of 105,500 square kilometers with a linear distance of approximately 450 km from east to west and north to south and accounts for 1.06% of China's total land area. It is one of the smallest provinces in terms of area. The topography of Zhejiang is characterized by mountains and hills (70.4%), plains and basins (23.2%), and rivers and lakes (6.4%). The arable land covers only 2.08 million hectares, which results in high habitat complexity. The terrain of Zhejiang slopes in a stepped manner from southwest to northeast. The southwestern part is dominated by mountains, the central part by hills, and the northeastern part is characterized by a low-lying alluvial plain. It can be roughly divided into six topographical regions: North Zhejiang Plain, West Zhejiang Hills, East Zhejiang Hills, Central Jinqu Basin, South Zhejiang Mountains, and Southeast Coastal Plain with Islands. Zhejiang has a subtropical monsoon climate characterized by distinct seasons, moderate annual temperatures, abundant sunlight, plentiful rainfall, high humidity, and synchronous changes in the rainy and hot seasons. The annual average temperature ranges from 15 to 18 °C, with extreme maximum temperatures reaching 44.1 °C and extreme minimum temperatures dropping to −17.4 °C. The annual average rainfall in Zhejiang ranges from 980 to 2000 mm, with an annual average sunshine duration of 1710 to 2100 h. Due to the influence of the ocean, Zhejiang enjoys superior temperature and humidity conditions compared to the inland monsoon regions at the same latitude, making it one of the few regions with relatively favorable natural conditions in China.

The forest area in Zhejiang is 6.68 million hectares, with a forest coverage rate of 60.5%. The total standing volume of living trees is 194 million cubic meters. Zhejiang Province is rich in vegetation resources, with over 3000 species, and includes 45 species of wild plants under national protection. The province is known for its rich tree species and is often referred to as the "Southeast Plant Treasure Trove." Zhejiang has nearly 4000 species of vascular plants, including over 1300 species of woody plants that belong to 109 families and 423 genera. This includes 8 families and approximately 45 species of gymnosperms, as well as 101 families and over 1260 species of angiosperms, many of which are unique to China.

### 2.2. Sample Site Setting and Investigation

We primarily selected ecological monitoring plots for these forest plots that were dominated by *Schima superba*, with a chronological age of 10 years or more and undisturbed

stands, in Zhejiang Province from 2011 to 2021 as samples. The plots for the Schima superba community are distributed across 20 counties in six cities in Zhejiang Province that cover various topographies, such as mountains, low hills, hills, and basins. The plots include uphill, mid-slope, and downhill locations with distribution across an elevation span of 0 to 1900 m, which ensures the rationality and reliability of plot distribution. The plots are distributed in Hangzhou City (Chun'an, Fuyang, and Jiande, totaling 125 plots), Huzhou City (Changxing, 2 plots), Jinhua City (Panan, Wuyi, Yiwu, and Yongkang, totaling 168 plots), Lishui City (Jingning, Jinyun, Longquan, Qingyuan, Songyang, and Suichang, totaling 465 plots), Quzhou City (Changshan, Jiangshan, Kaihua, Kecheng, and Qujiang, totaling 23 plots), and Taizhou City (Xianju, 48 plots), and they total 831 forest dynamic monitoring plots with a size of 20 m × 20 m (Figure 1). The altitude range of the sample plots was from 20 to 2000 m.

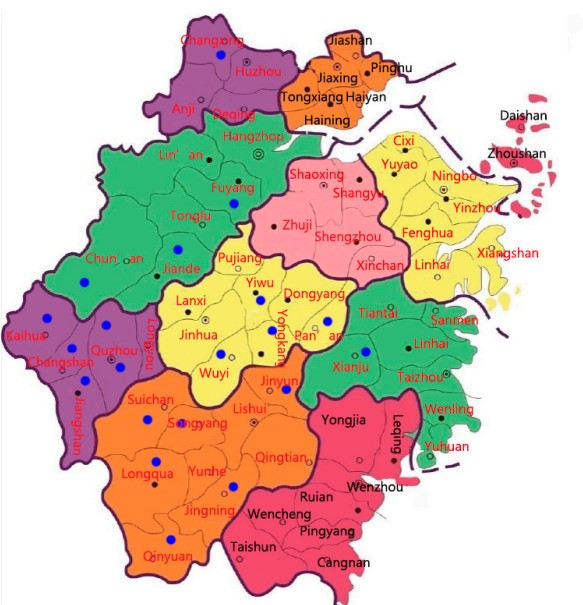

**Figure 1.** Distribution of study area. Note: the blue circles are the study sites, hollow circles represent the county area, and solid circles represent the city area. The areas without sample plots were mostly sites with relatively little cultivation and growth of *Schima superba*.

For each survey plot, we recorded information on surviving woody plants with a diameter at breast height (DBH) ≥ 1 cm and included species name, DBH, tree height, coordinates, branching status, and sprouting condition. Ecological factors such as geographical location, topography, soil texture, soil type, elevation, slope, aspect, slope position, community structure, vegetation cover, soil layer thickness, and litter thickness were also documented. The community structure is entirely composed of evergreen broad-leaved forests, with the dominant species being *Schima superba* and other species including *Quercus glauca* and *Liquidambar formosana*.

### 2.3. Sampling and Analysis of Environmental Factors

#### 2.3.1. Acquisition of Meteorological Factors

Meteorological data were primarily obtained through two sources. First, data for the respective regions of each plot were acquired from meteorological stations and ecological positioning observation research stations. These data included annual average temperature, annual precipitation, relative humidity, and minimum temperature. Second, temperature and humidity data were collected using temperature and humidity loggers (HOBO U23-002, Guangzhou, Guangzhou Junchong Electronic Technology Co., Ltd., Guangzhou, China) and small weather instruments installed at the 831 plots [19].

2.3.2. Collection and Treatment of Soil Factors

Soil samples were collected during the period from July 2011 to 2021 (when forestry survey fieldwork was conducted). In each of the 831 forest survey plots, the upper layer of litter was removed, and five random soil samples (0–20 cm deep) were collected using a soil auger. A "quartering method" was employed to retain approximately 1000 g of well-mixed soil samples, which were bulked to create a composite sample for each plot. After initial processing, the soil samples were analyzed for soil pH, soil organic matter (SOM), total nitrogen (TN), total phosphorus (TP), available nitrogen (AN), available phosphorus (AP), and available potassium (AK). The elevation, slope, litter thickness, soil thickness, humus layer thickness, and vegetation cover were recorded as observed values. The slope aspect was represented by numerical levels: the upslope was level 1, the mid-slope was level 2, and the downslope was level 3. Slope direction was graded numerically with north as level 1, northeast and northwest as levels 2 and 3, east and west as levels 4 and 5, southeast and southwest as levels 6 and 7, and south as level 8. Soil type and soil texture were also classified accordingly.

*2.4. Data Analysis*

2.4.1. Selection of Environmental Variables

We determined the percent contribution (PC) by continuously adjusting the coefficients of individual factors and providing the contribution of a specific climate variable to the species in percentage form. Permutation importance (PI) determines the values of environmental variables by randomly permuting the training point set and presenting them as percentages after normalization. The larger the percent contribution and permutation importance values of a climate variable, the more crucial it is to the potential geographic distribution of the species [3,14]. To select key environmental variables that affect the distribution of Schima superba, all 22 environmental variables were initially added to the MaxEnt model (Table 1). The contribution of each variable to the prediction results was analyzed. Subsequently, we used the band collection statistics tool in ArcGIS 10.8 software to calculate the correlation between environmental variables. In cases where there was high collinearity ($|r| \geq 0.8$) among variables, the variable with the higher contribution was retained.

**Table 1.** The contribution rate of the 22 environmental factors.

| Environmental Factor | Specific Name | PC (%) |
|---|---|---|
| elevation | Elevation | 37.6 |
| bio13 | The wettest monthly precipitation | 12.8 |
| bio19 | The cold season precipitation | 9.7 |
| bio2 | Mean diurnal temperature range | 4.4 |
| bio16 | Rainfall in the wettest season | 4.2 |
| bio7 | Temperature Annual Range | 4.2 |
| bio12 | Annual precipitation | 4.1 |
| bio15 | Seasonal variation in precipitation | 3 |
| bio6 | The lowest temperature in the coldest month | 2.6 |
| bio3 | Isothermality | 2.4 |
| bio14 | Driest Month Precipitation | 2.3 |
| bio17 | Driest Season Precipitation | 2.2 |
| aspect | Aspect | 2.2 |
| bio10 | Average air temperature in the hottest season | 1.7 |
| bio4 | Seasonal changes in air temperature | 1.6 |
| bio11 | Average air temperature in the coldest season | 1.3 |
| bio8 | Average air temperature in the wettest season | 0.9 |
| bio18 | Hottest Season Precipitation | 0.8 |
| slope | Slope | 0.7 |
| bio5 | Peak temperature in the hottest month | 0.7 |
| bio1 | Annual average temperature | 0.5 |
| bio9 | Average air temperature in the driest season | 0.2 |

Note: PC represents the contribution rate.

### 2.4.2. Model Accuracy Determination

The MaxEnt model uses the area under the receiver operating characteristic curves (AUC) (the receiver operating characteristic curves, ROC curves) to judge the accuracy of the model predictions. The AUC value is a crucial indicator to evaluate the reliability of the prediction results. AUC values close to 1 indicate a strong correlation between environmental variables and distribution models, which indicates higher prediction accuracy. Generally, $0.5 \leq AUC < 0.7$ suggests moderate predictive ability, $0.7 \leq AUC < 0.9$ suggests good predictive ability, and $0.9 \leq AUC < 1$ indicates excellent predictive ability and reflects the potential range of species distribution with high precision.

### 2.4.3. Prediction of Suitable Habitat

We filtered the distribution points for Schima superba and data for 22 climate variables and imported them into MaxEnt 3.4.1 [14]. A random selection of 75% of the distribution points was used as the training set, while 25% was used as the test set. Cross-validation was performed ten times with a maximum background point quantity of 10,000 and a maximum iteration number of 500. Other default settings were not changed. The final output was an ASCII format file. The average ASCII format files were then selected, imported into ArcGIS 10.5 software [4,14], and subjected to reclassification using the Reclassify command with the Spatial Analyst Tools. Jenks' natural breaks were applied to classify the suitability levels. The suitability values obtained from the MaxEnt model ranged from 0 to 1 and were divided into four levels based on suitability: suitability value < 0.05 indicates non-suitable areas where growth is not favorable, $0.05 \leq$ suitability value < 0.25 indicates low-suitability areas where growth is possible but not optimal, $0.25 \leq$ suitability value < 0.5 indicates moderately suitable areas where growth is relatively favorable, and suitability value $\geq 0.5$ indicates highly suitable areas where Schima superba was most likely to thrive.

### 2.4.4. Analysis of Bioclimatic Characteristics in Suitable Zones

The Jackknife test sequentially [4,14] uses and excludes a specific variable to build new models. The differences in Regularized Training Gain, Test Gain, and AUC values between models are compared to measure the importance of bioclimatic variables. The Jackknife test is used to detect the contribution of each environmental factor to the prediction results. Higher Regularized Training Gain values for "only including this variable" indicate greater predictive accuracy and a larger contribution to predicting species distribution. A greater reduction in Regularized Training Gain values for "excluding this variable" compared to "all variables" indicates that the variable contains more unique information and is more important to predict species distribution. The contribution rate, permutation importance values, and Jackknife method were used to evaluate the dominant climate variables that influence species distribution and analyze the importance of bioclimatic variables in restricting the modern geographic distribution pattern of Schima superba. Generally, when the distribution probability value was >0.5, the corresponding ecological factor values were suitable for plant growth.

### 2.4.5. Data Analysis and Graphical Processing

The distribution prediction of potential suitable zones for Schima superba was performed using MaxEnt 3.4.4 software. ArcGIS 10.2 software was used to process distribution maps for suitability levels and calculate the area of each suitable zone. Data were organized using Excel. Principal component analysis (PCA) was conducted using the "factoextra" package in R 3.2.5 software [14].

## 3. Result

### 3.1. Environmental Factor Screening

Based on the contribution of each variable to the prediction and correlation results in the MaxEnt model (Table 1), we selected a total of 22 variables for subsequent predictions: elevation, slope, aspect, mean diurnal temperature range (bio2), isothermality (bio3),

minimum temperature of the coldest month (bio6), temperature annual range (bio7), mean temperature of the wettest quarter (bio8), mean temperature of the driest quarter (bio9), precipitation of the wettest month (bio13), precipitation of the driest month (bio14), and precipitation of the driest quarter (bio17).

### 3.2. Model Accuracy Description

We set the number of repetitions for the MaxEnt model to 15 and constructed 15 potential distribution models with the average taken as the final prediction result. The average AUC value for repeated experiments was 0.804, with a standard deviation of 0.014, which indicates that the predictions were highly reliable (Figure 2).

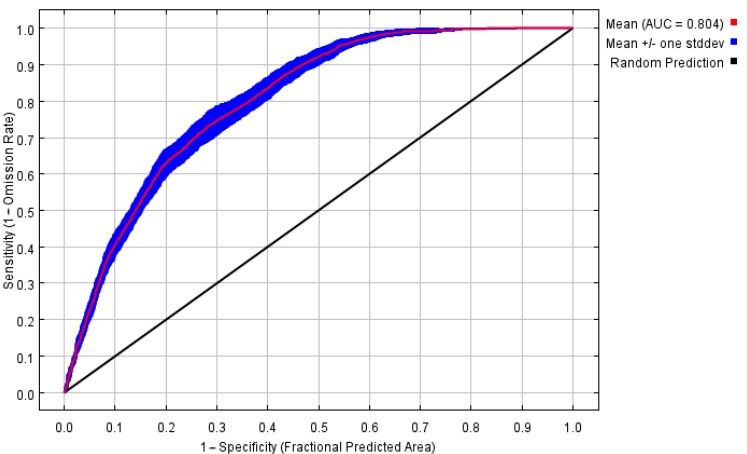

**Figure 2.** ROC curve of the wood load based on MaxEnt model operation.

### 3.3. Prediction Results of Suitable Areas

Figure 3 illustrates the final suitability habitat map for *Schima superba* in Zhejiang. The total suitable habitat area (suitability value > 0.05) is 87,600 square kilometers and possesses highly suitable areas that cover 29,400 square kilometers, moderately suitable areas that cover 25,700 square kilometers, and low-suitability areas that cover 32,500 square kilometers.

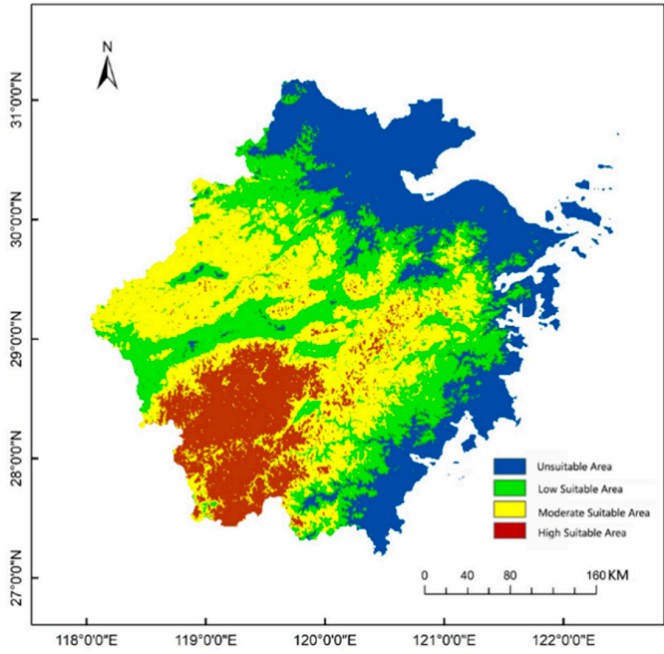

**Figure 3.** Analysis of suitable areas in Zhejiang Province based on the MaxEnt model.

### 3.4. Analysis of Climatic Characteristics for Ecological Fitness Organisms

The Jackknife method shows that precipitation during the most humid month (bio13) and elevation have a significant impact on the distribution of *Schima superba* (Figure 4). Likewise, precipitation during the driest quarter (bio17) and temperature annual range (bio7) contained a considerable amount of unique information not duplicated by other variables, which suggested that their distinctiveness influences *Schima superba* habitat distribution (Table 2).

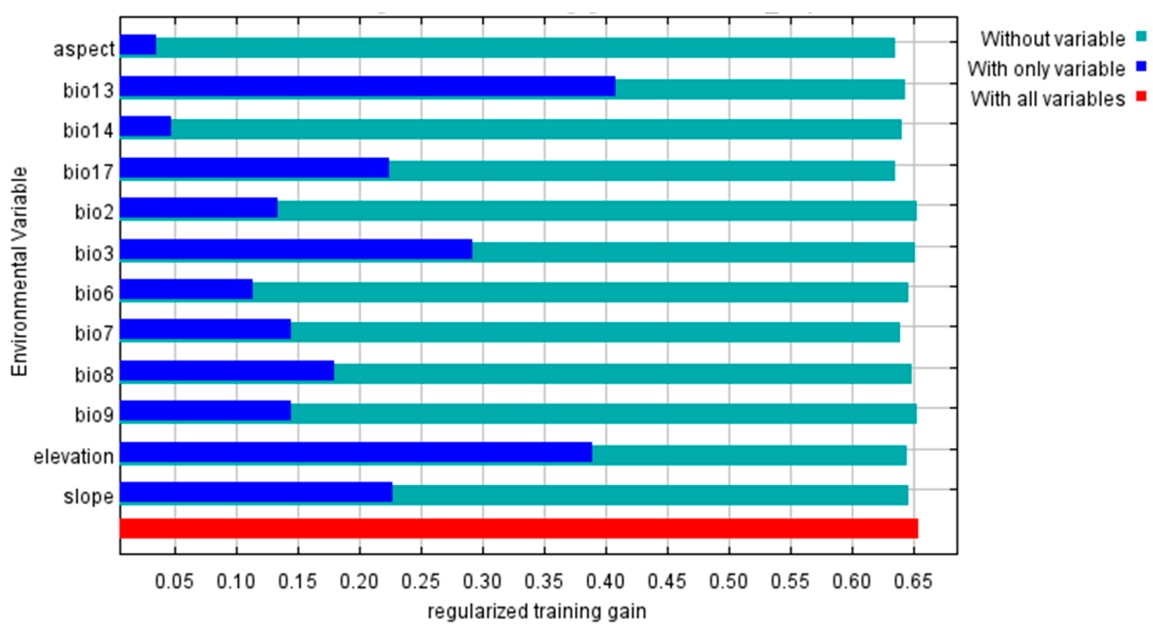

**Figure 4.** The Jackknife method assesses the importance of bioclimatic variables on distribution gain.

**Table 2.** Percent contribution of the 12 climatic features to the suitable habitat of *Schima superba*.

| Environmental Factor | Specific Name | PC (%) |
|---|---|---|
| elevation | Elevation | 46.7 |
| bio13 | The wettest monthly precipitation | 19.4 |
| bio17 | Driest Season Precipitation | 6.2 |
| bio3 | Isothermality | 5.3 |
| bio6 | The lowest temperature in the coldest month | 5.1 |
| bio2 | Mean diurnal temperature range | 4.9 |
| bio14 | Driest Month Precipitation | 4.2 |
| bio7 | Temperature Annual Range | 3.1 |
| bio8 | Average air temperature in the wettest season | 1.8 |
| aspect | Aspect | 1.7 |
| slope | Slope | 1 |
| bio9 | Average air temperature in the driest season | 0.5 |

Note: PC represents the contribution rate.

Figure 5 illustrates the response curves of *Schima superba* distribution to individual environmental variables using a single environmental variable to build the MaxEnt model. The data showed that *Schima superba* mainly occurs in areas above 500 m in elevation. With the increase in elevation, the probability of *Schima superba* distribution initially increases and reaches its peak between 1200 and 1400 m. It then decreases with higher elevations. The distribution probability of *Schima superba* significantly increases with growth during the most humid month with high precipitation (between 300 and 350 mm). However, in areas with higher precipitation during the most humid month, the distribution probability slightly decreases. Likewise, the distribution probability was very low in regions where precipitation was less than 140 mm. Mainly, the distribution probability of *Schima superba*

first rose and then declined with the peak occurring between 160 and 170 mm during the hottest season. The temperature annual range also showed the same variable trend in distribution probability. Here, temperature annual range represents the difference between the highest temperature in the hottest month and the lowest temperature in the coldest month, and our results show the optimal range between 28 and 31 °C (Figure 5).

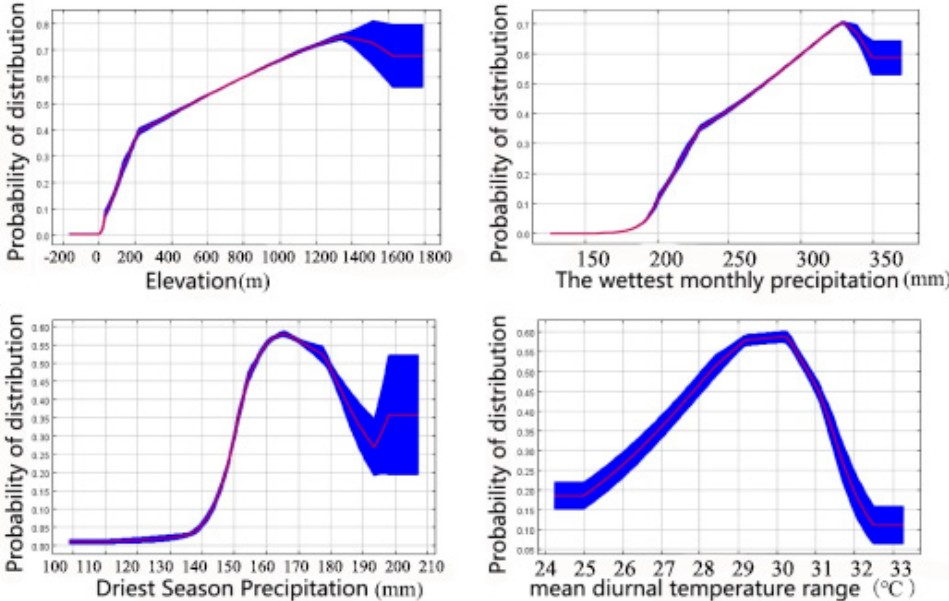

**Figure 5.** Response curves for *Schima superba* to the main environmental factors. The predicted interval values are shown in blue and the actual values are shown in red.

## 4. Discussion

Our results indicated that the adaptability of Schima superba in the Zhejiang Province was influenced by geographical distribution regions and environmental factors. Based on our contribution, permutation importance, and Jackknife test results obtained from the MaxEnt model simulation, we found that elevation, the most humid month's precipitation, and temperature annual range are the dominant climatic variables that affect the potential geographical distribution of Schima superba. In other words, the most critical factors that shape the Schima superba distribution pattern are elevation, precipitation, and temperature. Yao et al. found that Schima superba tends to be clustered and distributed on higher altitude slopes in subtropical secondary evergreen broad-leaved forests, with large-diameter individuals mainly concentrated on the uphill positions [19].

Previous research has found that the distribution and spatial variation of Schima superba in China is also influenced by precipitation and temperature, with precipitation playing a relatively more significant role in vegetation dynamics than temperature [20]. In terms of latitudinal characteristics, the adaptability of Schima superba to change in environmental factors decreases from east to west, and precipitation is identified as the primary factor that contributes to its weaker adaptability. However, this tends to underscore the importance of precipitation in the suitability for growth and geographical distribution of Schima superba [21–23]. This may be because Schima superba has the highest precipitation utilization efficiency in the understory in evergreen broad-leaved forests [19]. Therefore, precipitation plays a crucial role in the potential geographical distribution and growth of the Schima superba. However, it is important to note that temperature should not be overlooked, because it plays an irreplaceable role in constraining Schima superba's geographical distribution [24]. Our research results were consistent with those of another previous study, which confirms the accuracy and precision of the MaxEnt model [14]. Likewise, other studies have also shown that Schima superba species exhibit different morphological, ecological, and physiological characteristics in different geographical distribution regions,

which reflects their ability to adapt to the surrounding environment [25]. Regional differences, such as climate, soil types, elevation, and precipitation, have a significant impact on the growth and reproduction of Schima superba. For instance, in high-altitude areas, Schima superba may exhibit a shorter growth form to adapt to cold climate conditions and strong winds [21].

Climate is one of the key factors that influence *Schima superba* communities [26,27]. Different species possess varying requirements for climate factors, such as temperature, humidity, and precipitation [28]. Some species prefer a warm and humid climate, while others can adapt to colder or drier environments [23,24]. Changes in climate factors can affect the growth, flowering, and reproductive timing [29] of *Schima superba*. Likewise, soil conditions are crucial for the growth and adaptability of *Schima superba* communities [18,29]. Ultimately, *Schima superba* species have different requirements for soil texture, drainage, pH, and nutrient content, and adequate moisture is one of the key limiting factors for *Schima superba* communities. Sufficient moisture is essential for their growth and survival [29]. Additionally, light conditions play an important role in their growth and reproduction. For instance, adequate sunlight favors their growth and flowering [16]. Within forests, the canopy shade limits sunlight availability, so different species have varying needs and tolerances for light [17]. In summary, environmental factors play significant roles in the distribution, growth, and reproduction of *Schima superba* communities [6,16], and understanding their impact contributes to better conservation and management to maintain their biodiversity and ecosystem functions.

In addition to the factors we studied, other factors that contribute to species spatial distribution include ultraviolet radiation, interspecific competition, human activities, and species-specific characteristics [18]. Together, these factors are complex and diverse and influence species distribution in not well-understood ways, which will require more research to comprehensively characterize changes to species habit distributions. Altogether, future research would benefit from incorporating the mentioned variables with Schima superba to improve the accuracy of the predictive models. Nevertheless, our results serve as the first step in conservation planning efforts and provide a reference for future measures in Schima superba cultivation [19].

## 5. Conclusions

Our analysis using the MaxEnt model indicates that elevation, precipitation, and temperature are the primary climatic variables that influence the distribution of *Schima superba*. The suitable habitat total area for *Schima superba* in Zhejiang Province (suitability value > 0.05) is 87,600 km$^2$, with high-suitability, moderate-suitability, and low-suitability areas covering 29,400 km$^2$, 25,700 km$^2$, and 32,500 km$^2$, respectively. *Schima superba* predominantly occurs in areas with an elevation above 500 m, and the probability of its distribution reaches its peak between 1200 and 1400 m. The distribution probability of *Schima superba* significantly increased due to growth during the most humid month with high precipitation (300–350 mm). In areas where precipitation less was than 140 mm during the hottest season, *Schima superba* was largely absent. However, its distribution probability peaked when precipitation increased to 160–170 mm. With these results, the distribution probability of *Schima superba* tends to increase with an increase in temperature and peaks between 28 and 31 °C. Therefore, environmental factors' change significantly affects the suitable habitat area of *Schima superba*. Ultimately, climate change effects could be mitigated by planting more *Schima superba* in these optimal habitat areas, and it would increase carbon targets in southern China.

**Author Contributions:** L.Y. wrote the first draft of the manuscript and performed the data analysis. B.J. designed this study and improved the English language and grammatical editing. J.J. and C.W. conducted the fieldwork. The data support was provided by Y.X. and J.F. All the coauthors contributed to the discussion, revision, and improvement of the manuscript. All authors have read and agreed to the published version of the manuscript.

**Funding:** This research was financially supported by the Zhejiang Provincial Natural Science Foundation project (LQ23C030001), Zhejiang Provincial Scientific Research Institute special project (2022F1068-2); "Pioneer" and "Leading Goose" R&D Program of Zhejiang (2022C02053); and the Major Collaborative Project between Zhejiang Province and the Chinese Academy of Forestry (2021SY08).

**Data Availability Statement:** No new data were created or analyzed in this study. Data sharing is not applicable to this article.

**Acknowledgments:** We would like to thank Mr. Savannah Grace at the University of Florida for her assistance with the English language and grammatical editing of the manuscript.

**Conflicts of Interest:** The authors declare no conflict of interest.

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
