# Peer review of "Adaptability Analysis of the Evergreen Pioneer Tree Species Schima superba to Climate Change in Zhejiang Province"

_forests, doi:10.3390/f14122438_

Round 1
Reviewer 1 Report (Previous Reviewer 3)
Comments and Suggestions for Authors
The publication has been revised in accordance with my suggestions. The methodology describes in more detail the collection and laboratory analysis of soil samples. The discussion compared our own results with those of others concerning the same species, not, for example, rice. The publication is suitable for further processing.
Author Response
Please see the attachment.

Reviewer 2 Report (Previous Reviewer 2)
Comments and Suggestions for Authors
The authors have improved the paper considerably. I don't have any further comments.
Author Response
Please see the attachment.

Reviewer 3 Report (Previous Reviewer 1)
Comments and Suggestions for Authors
The manuscript was improved, but it is still not clear and well structured.
The abstract
- must have the concept, M&M, results, and conclusions. By this order. Concept must be written in present, all other parts in past. Do not mixture the order.
Introduction
- talks about climate changes and the whole text is talking, but you did not work any simulation of climate change, not explore the extreme events. You CANNOT use climate change in your work.
- Introduction must talk about the species of interest and what it needs of environment.
- Describe generally models and after that specifically models to be used.
M&M
- Too much text about regions, but they must be classified somehow by environment
- Too much about the forest, but no type of forest explained, which are the dominant species, density, height…. where is the species of interest, what %, age, dynamics.
- The description of plots was improved, but it is still poor.
- Data collected in different localizations in a temporal distance?
- The environmental data can simply be mentioned in M&M, you do not need the Table 1. BUT the frequency of datalogger register, sensors for measurements etc. MUST be done in details. No information about that!
- How did you determine community structure? What were the other species?
Results
- Give the explanation of PC% under the Table.
- Write impersonally, not that table or figure shows something.
Discussion
- Still immature. Compare your results to some other publications, make the synthesis of your work and projections too.
Conclusions are more discussion than proper discussion.
Use of references: Always put some reference(s) when you are talking about the knowledge that someone described.
Always write about your results using the verbs in past and about the published results, yours or of other scientists, in present.
Details are done in pdf file.

OK, BUT SOME WORDS, AS 'REALIZE', HAD NOT BEEN APPLIED IN A GOOD MEANING.
Round 2
Reviewer 3 Report (Previous Reviewer 1)
Comments and Suggestions for Authors
The manuscript was significantly improved. Some details as the upper case letters used to indicate one abbreviation (not necessary), or use the times of verbs must be revised. I indicated some in the attached manuscript.

Author Response
Dear reviewer,
According to your comments, we have revised the inappropriate verbs, adverbs and case and so on in the article, please review.
Thank you very much for your consideration.
Yours sincerely,
Liangjin Yao
This manuscript is a resubmission of an earlier submission. The following is a list of the peer review reports and author responses from that submission.
Round 1
Reviewer 1 Report
Comments and Suggestions for Authors
The manuscript use the environmental factors that had not been defined until results
Introduction talks about all, but not about what was done.
M&M starts with regions, but never explain no method, number, position of plots, how many trees were analyzed… When and where the data were collected . Can you compare data collected in different localizations in a temporal distance of six months?
You did not explain no environmental data, no measurement… biomass???
How did you determine community structure? What were the other species?
Some methods had not been described in M&M but in Results.
Rewrite, recalculate, restructure. Be scientific, some long stories are for local public, not international scientific podium.
· Always write about your results using the verbs in past and about the published results, yours or of other scientists, in present.
· Details are done in pdf file.

OK
Reviewer 2 Report
Comments and Suggestions for Authors
Understanding the habitat adaptation of plants is of great significance for the protection and management of natural ecosystems, as well as predicting species distribution and responses to climate change. Therefore, the topic of paper is relevant.
The paper is well written. I don't have any serious comments. Typos need to be corrected.
The scientific novelty lies in analysis of environmental factors and habitat suitability prediction based on survey data of Schima superba and habitat factors from 831 fixed forest monitoring plots in 20 counties (cities) in Zhejiang Province.
The paper has both theoretical and practical significance. This study reveals the ecological characteristics and adaptation mechanisms of Schima superba in different geographical regions of Zhejiang Province, providing a scientific basis for the protection and sustainable management of Schima superba resources in response to climate change.
In the introduction, the relevance of the study is well substantiated and the state of the problem is described. The research objectives are formulated clearly and clearly.
The research area is described well. This study was primarily conducted in 20 counties (cities) in six cities of Zhejiang Province.
The methodology is described in detail. The authors used modern methods of analysis. The choice of methods is reasonable and adequate for the tasks set.
The research results are illustrated with figures and tables that are informative and do not duplicate each other. The paper contains 2 informative tables and 5 visual figure. The results are presented clearly and clearly.
Conclusions follow from the results and are reasonable. The article will be of interest to a wide range of readers whose scientific interests are related to ecology, as well as climate change. Despite the fact that English is not my native language, I read the paper with interest and had no difficulties in understanding. The paper fully corresponds to the subject and scientific level of the Forests.
Reviewer 3 Report
Comments and Suggestions for Authors
Interesting research, in a large number of research sites, illustrating the adaptation of the studied species to changing climatic factors. There are numerous editorial errors throughout the publication. In particular, the description of laboratory methods for soil assessment and other research methods used, indicated in the publication, should be improved. The discussion should also be expanded to include the results of research by other authors on the adaptation of woody plants to variable climatic factors, this was written in very general terms in the discussion.

The publication is written correctly in English.